# Influenza A virus NS1 protein hijacks YAP/TAZ to suppress TLR3-mediated innate immune response

Qiong Zhang[1], Xujun Zhang[1], Xiaobo Lei 🔵[2]*, Hai Wang[3], Jingjing Jiang[1], Yuchong Wang[1], Kefan Bi[1], Hongyan Diao🔵[1]*

**1** State Key Laboratory for Diagnosis and Treatment of Infectious Diseases, National Clinical Research Center for Infectious Diseases, Collaborative Innovation Center for Diagnosis and Treatment of Infectious Diseases, The First Affiliated Hospital, College of Medicine, Zhejiang University, Hangzhou, Zhejiang, China, **2** NHC Key Laboratory of System Biology of Pathogens, Institute of Pathogen Biology, Chinese Academy of Medical Sciences & Peking Union Medical College, Beijing, P.R. China, **3** Department of Laboratory, Tongde Hospital of Zhejiang Province, Hangzhou, Zhejiang, China

* fyleixb@126.com (XL); diaohy@zju.edu.cn (HD)

**Data Availability Statement:** All relevant data are within the manuscript and its Supporting Information files.

## Abstract

The Hippo signaling pathway, which is historically considered as a dominator of organ development and homeostasis has recently been implicated as an immune regulator. However, its role in host defense against influenza A virus (IAV) has not been widely investigated. Here, we found that IAV could activate the Hippo effectors Yes-associated protein (YAP) and transcriptional coactivator with PDZ-binding motif (TAZ) through physical binding of the IAV non-structural protein 1 (NS1) with C-terminal domain of YAP/TAZ, facilitating their nuclear location. Meanwhile, YAP/TAZ downregulated the expression of pro-inflammatory and anti-viral cytokines against IAV infection, therefore benefiting virus replication and host cell apoptosis. A mouse model of IAV infection further demonstrated Yap deficiency protected mice against IAV infection, relieving lung injury. Mechanistically, YAP/TAZ blocked anti-viral innate immune signaling via downregulation of Toll-like receptor 3 (TLR3) expression. YAP directly bound to the putative TEADs binding site on the promoter region of TLR3. The elimination of acetylated histone H3 occupancy in the TLR3 promoter resulted in its transcriptional silence. Moreover, treatment of Trichostatin A, a histone deacetylases (HDACs) inhibitor or disruption of HDAC4/6 reversed the inhibition of TLR3 expression by YAP/TAZ, suggesting HDAC4/6 mediated the suppression function of YAP/TAZ. Taken together, we uncovered a novel immunomodulatory mechanism employed by IAV, where YAP/TAZ antagonize TLR3-mediated innate immunity.

## Author summary

The mechanisms of influenza A virus (IAV) infection, host immune responses and interplay of host cells and virus have been under intensive study for decades of years. This has largely improved our understanding on how human immune system responses against virus and how virus evolves and develops various strategies to evade host immune

**Funding:** H.D.was supported by funding from National Key Research and Development Program of China (2021YFA1301100, 2018YFC2000500), the Key Research & Developement Plan of Zhejiang Province (2019C04005), Research Project of Jinan Microecological Biomedicine Shandong Laboratory (JNL-2022012B). Q.Z. received funding from National Natural Science Foundation of China Grants (82101845). H.W. received funding from Public Welfare Foundation of Zhejiang Science and Technology Agency (LGF22H160047). The funders had no role in study, design, data collection and analysis, decision to publish, or preparation of the manuscript.

**Competing interests:** The authors have declared that no competing interests exist.

surveillance. However, the panorama is far from fully elucidated, and therapeutic strategies with higher specificity of IAV are still in urgent need. In this study, we uncovered a new strategy employed by IAV to mute host innate immune response, of which NS1, a multi-functional protein of IAV activates host proteins YAP/TAZ to antagonize TLR3 expression. TLR3 mediates important innate immune signaling that produces pro-inflammatory and anti-viral cytokines against infection, thus, loss of YAP/TAZ enhances host innate immune response and protects mice from lung injuries induced by IAV infection. Our study may provide a new potential target for prevention and treatment of IAV infection.

## Introduction

Influenza A virus (IAV) remains a major global public health threat due to its annual epidemics and pandemic potential. H1N1 is a most common subtype of IAV and accounts for higher morbidity and mortality every year [1,2,3]. IAV enters the host through nasal or oral cavities, and in most cases results in self-limited infection, contained to the respiratory tract [4]. Thus, epithelial cells in the respiratory tract are recognized to be primary targets for virus and contribute to both innate and adaptive immune function.

Pattern recognition receptors (PRRs) provided by the innate immune system are considered as formidable barriers to detect molecules of pathogens. Some PRRs, such as the retinoic acid-inducible gene I (RIG-I), the NOD-like receptor family member NOD-, LRR- and pyrin domain-containing 3 (NLRP3) and the Toll-like receptor members TLR3 and TLR7, have been implicated in viral RNA detection during IAV infection.

Activation of PRRs initiates downstream antiviral signaling and proinflammatory response [5,6,7,8,9,10]. Recognition of pathogen associated molecular patterns (PAMPs) triggers the recruitment of adaptor proteins which activate transcription factors interferon (IFN) regulatory factor 3 (IRF3), IRF7, and nuclear factor κB (NF-κB) and p38 mitogen-activated protein kinase [11,12,13,14].

Many pathogens, IAV included, have evolved various strategies to counteract cellular antiviral responses. The non-structural protein 1 (NS1) represents one of such viral proteins which can antagonize host innate immune response. This viral component has been shown to suppress type I IFN response through multiple tactics. NS1 inhibits RIG-I activation by degrading OTUB1, a critical regulator of RIG-I activation [14], or by interaction with TRIM25, which results in suppressed RIG-I ubiquitination and activation [15,16,17]. Recent studies also reported that NS1 inhibits NLRP3 inflammasome-mediated IL-1β secretion by interaction with NLRP3 and impairment of ASC speck formation and ubiquitination [18,19]. Despite these well described mechanisms, the full landscape of host anti-viral immune evasion by NS1 is far from fully elucidated.

The Hippo pathway has long been recognized as an important regulator of organ development and homeostasis [20,21]. The core components of the mammalian Hippo pathway are a kinase cascade involving mammalian STE20-like protein kinase 1 (MST1) and MST2; the large tumor suppressor 1 (LATS1) and LATS2; the adaptor proteins Salvador homolog 1 (SAV1) for MST1/2 and MOB kinase activators (MOB1A/MOB1B) for LATS1/2; downstream effectors Yes-associated protein (YAP) and its analog protein, transcriptional coactivator with PDZ-binding motif (TAZ); and TEA domain transcription factors (TEADs) [22,23,24,25,26]. When activated, Hippo signaling initiates a series of phosphorylation events via MST and LATS kinases, leading to the phosphorylation of YAP/TAZ, the key effectors of the pathway.

Phosphorylated YAP/TAZ are sequestered in the cytoplasm by 14-3-3 proteins, which inhibit their transcriptional activity and leads to polyubiquitination and degradation of YAP/TAZ in the proteasome [27,28]. By contrast, inactivation of the Hippo pathway facilitates dephosphorylation and nucleus accumulation of YAP/TAZ, which allows association with TEADs, and initiates transcription of target genes [20]. Recent reports illustrated that the innate antiviral immunity is also intensively regulated by the Hippo pathway [29,30]. Yet, the overall picture of the reciprocal regulation of the Hippo signaling and the innate antiviral response remains elusive.

In this study, we described that YAP/TAZ are over-activated upon IAV infection by viral multifunctional protein NS1. Activated YAP/TAZ in turn hamper TLR3-triggered innate immune signaling to facilitate evasion of host immune response for IAV infection, which eventually results in increased virus replication, cell death and severe lung injure. Our study provides functional and mechanistic insights into the reciprocal regulation of viral protein and the Hippo signaling during infection, which is beneficial for host immune evasion. Therefore, our study has important clinical impact for viral infection prevention and treatment.

## Results

### IAV infection activates YAP/TAZ

To investigate whether the Hippo signaling pathway is affected by influenza virus infection, human type II alveolar epithelial-like cell line A549 cells were infected with influenza virus strain A/PR8/1934 (PR8). Dephosphorylation and upregulation of the Hippo effector YAP/TAZ were observed (Fig 1A). Additionally, the transcription of CYR61 and CTGF, the target genes of YAP/TAZ, were increased upon infection of PR8 virus (Fig 1B). CTGF and GTIIC luciferase reporter assays also showed a boost of transcription activity of YAP/TAZ following PR8 virus infection, while mutated TEAD-binding sites on CTGF luciferase reporter construct (CTGF-ΔTB-Luc) was not activated, confirming the specificity of YAP/TAZ-TEADs transcriptional complex is required for the observed increase in luciferase activity (Fig 1C). Finally, the translocation and accumulation of YAP/TAZ in nucleus upon infection were observed in both immunofluorescence and subcellular fraction separation assays (Fig 1D and 1E). These results suggest that YAP/TAZ are activated upon IAV infection.

To confirm our *in vitro* observations, C57BL/6J mice were infected with the mouse-adapted PR8 virus intranasally to investigate the regulation of Yap/Taz activities *in vivo*. The pro-inflammatory cytokine Il-6 and antiviral type I interferon Ifnb1 were significantly up-regulated in the lung as early as 1-day post-infection, and gradually declined afterwards (S1A Fig). Notably, the activity of mouse Yap was dramatically stimulated in the lungs 2 to 4 days post-infection, which was accompanied by activation of Irf3 (Fig 1F). Besides, an increase of Ctgf and Cyr61 transcription was also detected (Fig 1G). Taken together, these results demonstrate that IAV promotes the activation of YAP/TAZ both *in vitro* and *in vivo*.

### IAV NS1 binds to YAP/TAZ to stimulate their activities

We then sought to determine which viral protein is responsible for the stimulation of YAP/TAZ activities. Since several literature has pointed out IAV NS1 could affect host cell signaling and control cellular processes such as activation of PI3K signaling [31,32], we hypothesized that that NS1 is responsible for the regulation of YAP/TAZ activities. Therefore, we transfected A549 cells with Flag tagged NS1 protein from PR8 virus, together with the luciferase reporter plasmids of CTGF or GTIIC. The results suggested that overexpression of NS1 potentiated the transcriptional activities of YAP/TAZ in a dose-dependent manner (Fig 2A). Consistently, NS1 overexpression dephosphorylated YAP, increased its protein level, and promoted the

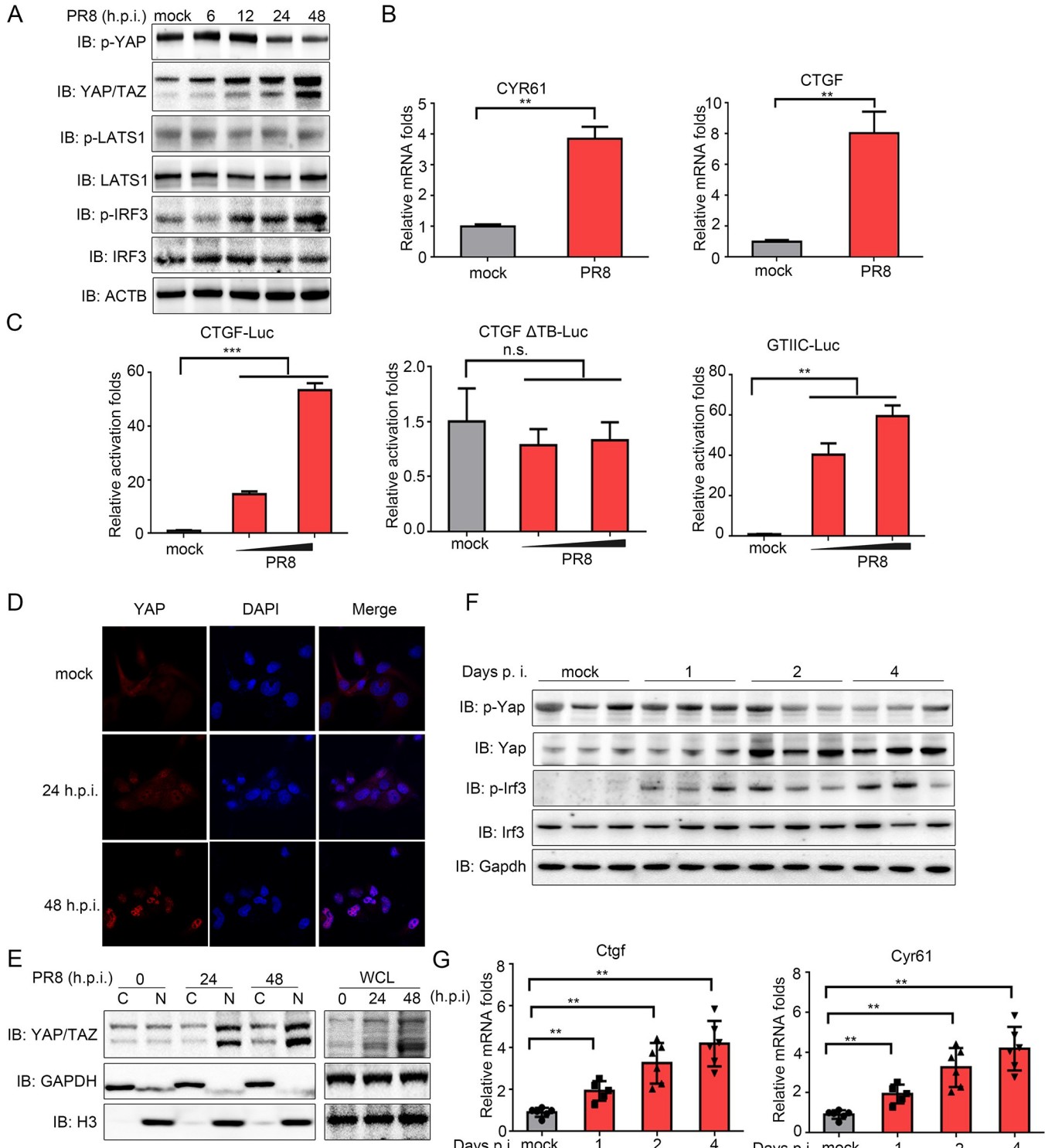

**Fig 1. IAV infection potentiates the activity of YAP/TAZ.** (A) A549 cells were infected with PR8 or mock virus at an MOI of 0.01 for 6, 12, 24 and 48 hours. Cells were then lysed and analyzed by SDS-PAGE. Immunoblotting was conducted with indicated specific antibodies. (B) A549 cells were infected with PR8 or mock virus at an MOI of 0.01 for 12 hours. Cells were then lysed and mRNA expression of CYR61 and CTGF was determined using RT-qPCR. (C) A549 cells were transfected with CTGF-Luc, CTGF ΔTB-Luc and GTIIC-Luc, along with Renilla luciferase control. At 12 hours post transfection, cells were either mock infected, or infected with 0.005 or 0.01 MOI of PR8 virus for another 12 hours, at which time point the promoter activities were measured using a dual-luciferase assay. (D) A549 cells were infected with PR8 or mock infected. Cells were fixed at 24 or 48 hours post infection and stained with anti-YAP antibody and DAPI for immunofluorescence microscopy. (E) Cytosolic (C) and nuclear (N) fractions and whole cell lysate (WCL) prepared from PR8-infected A549

cells were analyzed by SDS-PAGE and immunoblotting using antibodies specific for YAP/TAZ, GAPDH and Histone 3 (H3). (F) Female C57BL/6J mice were infected intranasally with $10^3$ PFU of PR8. Lungs were harvested at 1, 2 and 4 days post inoculation for immunoblotting. (G) Female C57BL/6J mice (n = 6 per group) were infected intranasally with PR8 or mock virus. Lungs were harvested for RT-qPCR. Data in (A-E) were repeated at least three times. Data in (B-C) and (G) are presented as means ± SD. $^*p < 0.05$, $^{**}p < 0.01$, $^{***}p < 0.001$. n.s., not significant ($p > 0.05$).

expression of its target gene CTGF (Fig 2B and 2C), which were congruent with our observations with PR8 virus infection (Fig 1A and 1B).

To elucidate the mechanisms by which NS1 elicits the activity of YAP/TAZ, we then tested for a potential direct interaction between NS1 and YAP/TAZ. Co-immunoprecipitation assay of exogenous NS1 protein of PR8 virus with exogenous YAP or TAZ in HEK293T cells showed a direct binding of NS1 to YAP/TAZ (Fig 2D). Furthermore, NS1 proteins of other IAV strains, including A/California/07/2009 and A/Hong Kong/156/1997, also bound YAP/TAZ (S2A Fig). Previous report suggested C-terminal transcriptional activation domain (TAD, residues 276–488) of YAP may be responsible for binding with other proteins to exert its function [33]. Indeed, we found YAP C-terminal TAD deletion (YAP del C) failed to immunoprecipitate with FLAG-NS1, suggesting C-terminal TAD of YAP was essential for its interaction with NS1 (Fig 2E).

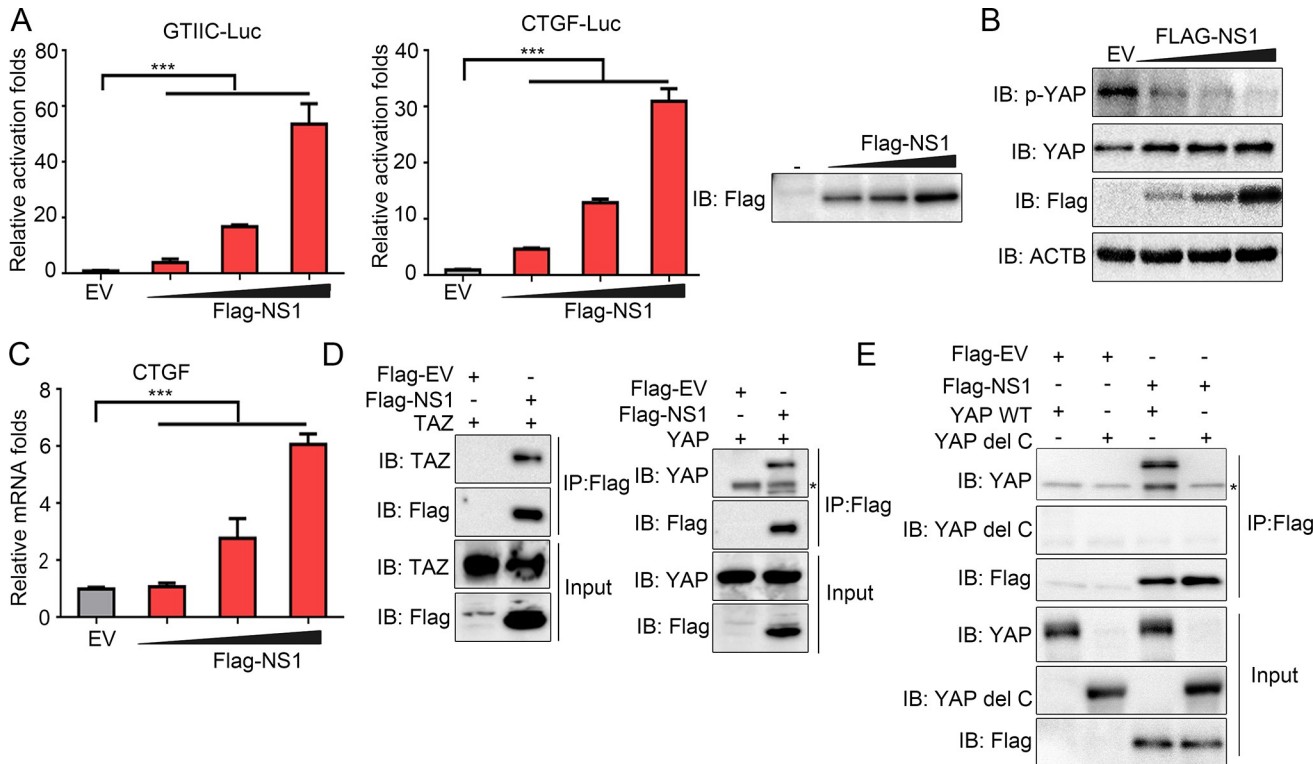

**Fig 2. NS1 induces YAP/TAZ activation.** (A) Luciferase reporter assays of GTIIC and CTGF reporter were conducted in A549 cells transfected with NS1 or EV plasmids for 24 hours. (B) A549 cells were transfected with different amounts of Flag-NS1 plasmid. WCL was analyzed by immunoblotting with indicated antibodies. (C) Flag-NS1 was transfected into A549 cells. WCL was subjected to RT-qPCR for CTGF expression. (D) 293T cells were co-transfected with TAZ or YAP plasmids, with Flag-NS1 and Flag-EV. WCL was subjected to immunoprecipitation with M2 beads, bound proteins and input lysis were analyzed by immunoblotting with indicated antibodies. $^*$, nonspecific bands. (E) YAP WT or YAP del C plasmids were co-transfected with Flag-NS1 or Flag-EV into HEK293T cells. WCL was subjected to immunoprecipitation with M2 beads, followed by immunoblotting. $^*$, nonspecific bands. All experiments were repeated at least three times. Data in (A) and (C) are presented as means ± SD. $^*p < 0.05$, $^{**}p < 0.01$, $^{***}p < 0.001$.

## YAP/TAZ promote replication and pathogenicity of IAV

The above data suggest that NS1 could regulate the activities of YAP/TAZ. Thus, we subsequently determined whether the Hippo signaling controls the antiviral response to IAV. A549 cells, stably expressing YAP(5SA), the constitutively active form of YAP, and LATS1 were generated (S3A Fig), which were then infected with PR8-GFP virus. The virus proliferation and replication can be detected and quantified by measurement of fluorescence in cells. Compared to empty vector (EV)-overexpressing cells, virus replication in YAP(5SA)-overexpressing cells was substantially enhanced, but decreased in LATS1-overexpressing cells (Fig 3A, left and upper right panels). The viral titers were also regulated in the same way (Fig 3A, lower right panel). Accordingly, compared to control cells, knockdown of YAP/TAZ restricted the replication of PR8, while LATS1/2 knockdown facilitated its replication (Figs 3B, S3B and S3C). The well-known IAV sensor RIG-I was also knocked down, which resulted in a significantly enhanced in GFP fluorescence and viral titers (Figs 3B and S3E). These results were also validated by measurement of viral copies (S4A and S4B Fig).

IAV infection can induce apoptosis. Compared to EV-overexpressed cells, the apoptosis rate induced by IAV infection in YAP(5SA)-overexpressing cells increased, while in LATS1-overexpressing cells decreased (Fig 3C).

*In vivo*, genetic deletion of Yap1 in mice results in early embryonic death [34]. Thus, in order to assess the impact of YAP in IAV proliferation and pathogenicity *in vivo*, we generated heterozygous Yap$^{+/-}$ mice using CRISPR/Cas9 (S4C Fig). Compared with wild-type (WT) mice, Yap$^{+/-}$ mice intranasally infected with PR8 virus showed lower viral burden as measured by viral mRNA expression and viral replication (Figs 3D and S4D). Furthermore, Yap deficiency resulted in less weight loss (Fig 3E). Histological analyses of infected lungs also revealed relieved lung injuries with milder alveolar collapse, less damaged pulmonary parenchymal architecture, and relieved focal interstitial thickening in Yap$^{+/-}$ mice compared to WT mice (Fig 3F, left panel). The apoptosis of parenchymal cells was determined by TUNEL/DAPI staining. Yap deficiency remarkably decreased the ratio of apoptotic parenchymal cells during infection (Fig 3F, right panel). Collectively, these results suggest that YAP/TAZ exacerbate virus proliferation and lung injury post infection.

## YAP/TAZ suppress IAV-induced innate immune response

Recent reports demonstrated YAP/TAZ negatively regulate innate immunity against many DNA and RNA viruses [35]. Thus, we sought to determine whether the Hippo signaling also regulates the innate immune response to influenza virus through the Hippo pathway components YAP/TAZ, TEADs and LATS1/2 knockdown with siRNAs in A549 cells (S5B–S5D Fig). These cells were then infected with PR8 virus to assess the impact of the Hippo pathway in the anti-viral response. Knockdown of YAP/TAZ or TEADs increased the transcriptional expression of antiviral and proinflammatory factors IFNB1, CXCL8 and IRF7 induced by virus infection. Meanwhile, knockdown of LATS1/2 decreased the transcriptional levels of these target genes (Fig 4A). Polyinosinic:polycytidylic acid (poly I:C) was used as a control stimulator. The results showed it had a similar magnitude of effect with PR8 virus upon siRNAs knockdown (Fig 4A). Consistently, cells stably expressing YAP(5SA) reduced the mRNA levels of IFNB1, IRF7 and CXCL8 following IAV infection (Fig 4B). In addition, overexpression of YAP effectively inhibited the IFNB promoter activation in a dose-dependent manner (Fig 4C, left panel), while overexpression of LATS1 induced it (Fig 4D). Notably, after deletion of C-terminal TAD, YAP was no longer able to suppress the IFNB promoter activation, indicating that C-terminal TAD was essential for YAP regulation of innate immune signaling (Fig 4C, right panel).

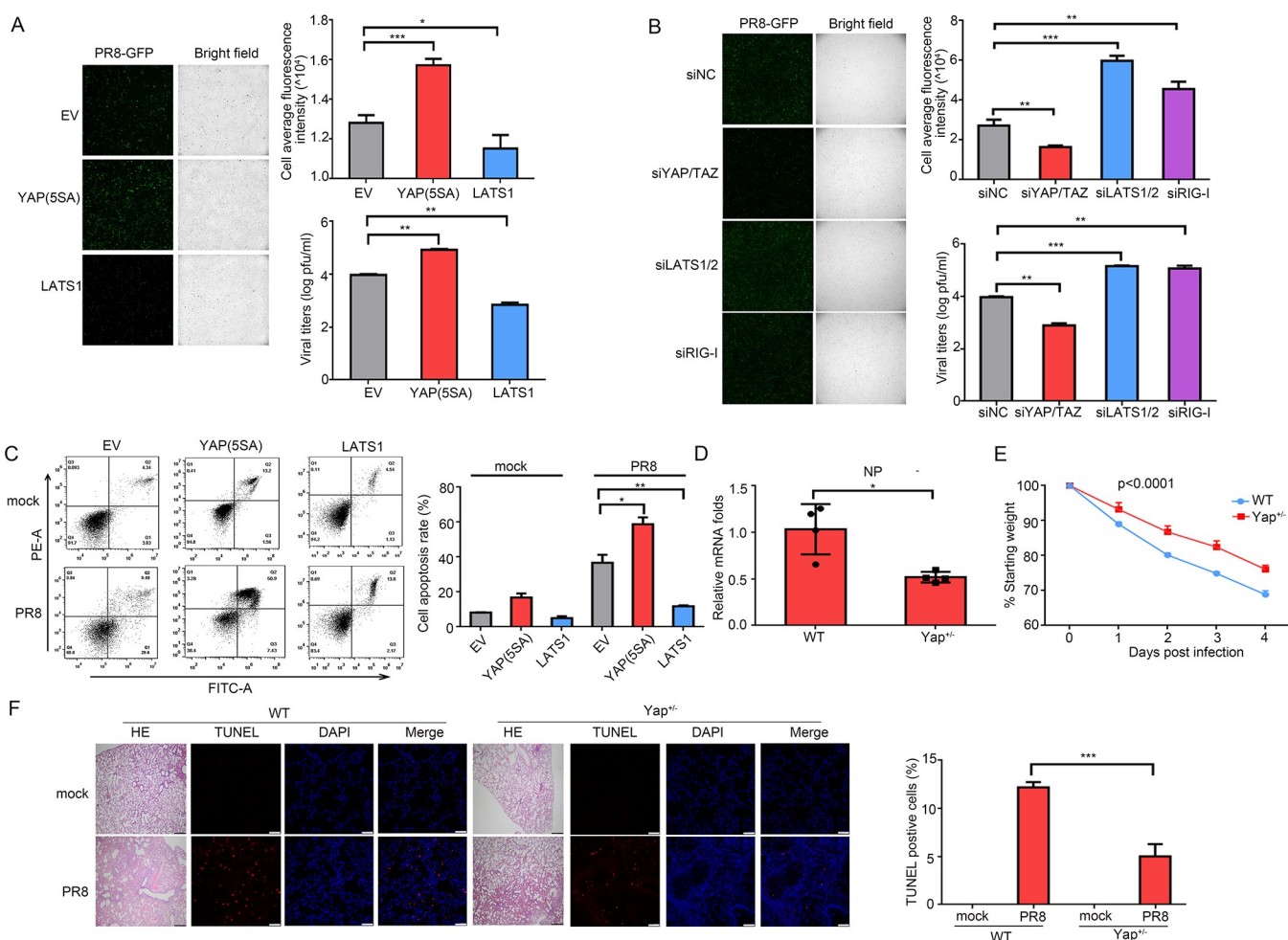

**Fig 3. YAP/TAZ antagonize host antiviral defense against IAV *in vitro* and *in vivo*.** (A, B) A549 cells either stably overexpressing EV, YAP(5SA) and LATS1 (A), or transfected with siNC, siYAP/TAZ, siLATS1/2 and siRIG-I (B) were infected with PR8-GFP at an MOI of 0.005. After 24 hours, GFP fluorescence was detected (left panel). Cell average fluorescence intensity per cell was calculated (upper right panel). The viral titers in the culture supernatants were measured using a plaque assay (lower right panel). (C) Cells overexpressing EV, YAP(5SA) and LATS1 respectively were infected with PR8 virus at an MOI of 0.02 for 24 hours. The apoptotic rate was analyzed by Annexin V-propidium iodide double staining. Representative plots of propidium iodide versus Annexin V-FITC fluorescence signals (left), and percentage of apoptotic cells (right) are shown. (D) WT and Yap$^{+/-}$ mice (n = 4 per group) were infected intranasally with $10^3$ PFU of PR8, and lungs were harvested for NP mRNA quantification by RT-qPCR. (E) Body weight loss (%) of WT and Yap$^{+/-}$ mice (n = 5 per group) after PR8 infection was measured. (F) Yap+/- and WT mice infected with PR8 and corresponding mock controls. The lung sections collected 2 days post infection were stained with haematoxylin and eosin. Scale bar: 200 μm. The apoptotic cells were showed by TUNEL staining with DAPI for cell nuclei (left panel). Scale bar: 40 μm. The percentage of TUNEL-positive cells was calculated (right panel). Data in (A-C) were repeated at least three times. Data in (A-F) are presented as means ± SD. $^*p < 0.05$, $^{**}p < 0.01$, $^{***}p < 0.001$.

To verify that those antiviral responses are also regulated by Yap *in vivo*, we measured the expression level of Ifnb1 and Irf3 in mock- or PR8-infected WT and Yap$^{+/-}$ mice. Yap$^{+/-}$ mice showed increased Ifnb1 and Irf3 expression in both mRNA and protein levels upon infection, which is in support of our *in vitro* results (Fig 4E and 4F). Altogether, these results suggest that YAP/TAZ attenuate antiviral innate immunity against IAV.

## YAP/TAZ hamper TLR3-mediated innate immune signaling pathway

PRRs sensing viral RNAs, such as RIG-I, TLR3 and TLR7, have emerged as critical intracellular receptors for the innate immunity against influenza infection [8,9,36]. Thus, we next addressed whether these PRRs are targets of the Hippo signaling during IAV infection. Knockdown of

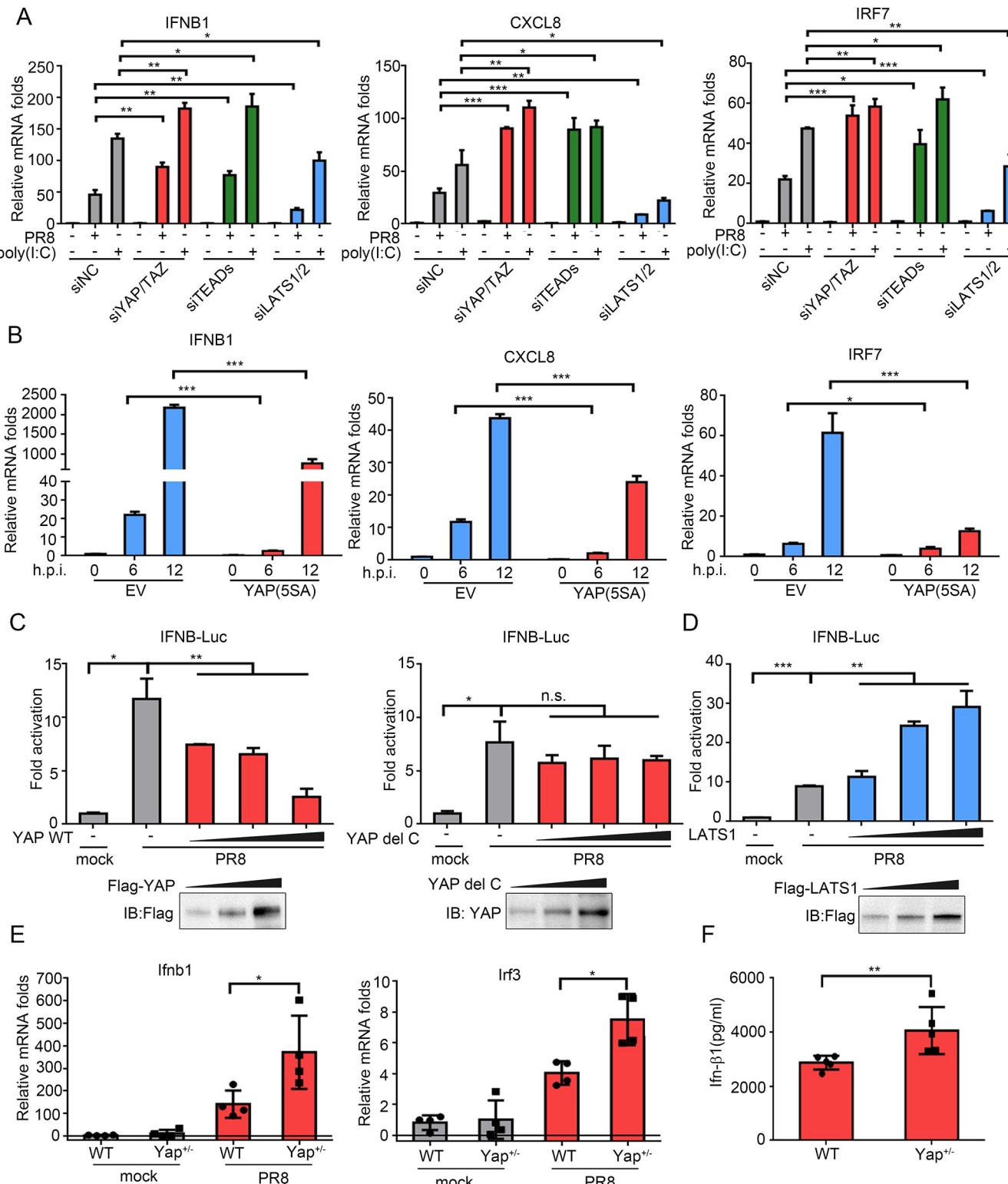

**Fig 4. Host innate immune response to IAV is regulated by YAP/TAZ.** (A, B) IFNB1, CXCL8, and IRF7 transcript levels were assessed by RT-qPCR in A549 cells with NC, YAP/TAZ, LATS1/2 and TEADs knockdown (A), or YAP(5SA) overexpression (B). Afterwards, cells were stimulated with PR8 virus or poly I:C. (C, D) The IFNB promoter activity in A549 cells co-transfected with different concentrations of YAP, YAP del C (C) or LATS1 (D) plasmids upon PR8 virus infection was measured by dual-luciferase assay. (E) Changes in mRNA expression of mouse Ifnb1 and Irf3 were assessed in the lungs of WT and Yap[+/-] mice (n = 4 per group) upon PR8 virus infection or corresponding mock controls. (F) Level of Ifn-β1 was assessed by ELISA in the lung tissues of WT

or Yap$^{+/-}$ mice (n = 5 per group). Data in (A-D) were repeated at least three times. Data in (A-F) are presented as means ± SD. $^*p < 0.05$, $^{**}p < 0.01$, $^{***}p < 0.001$. n.s., not significant (p > 0.05).

RIG-I and TLR3 significantly promoted virus replication. Whereas, knockdown of TLR7 barely showed any effect on virus replication (S5A, S3E and S3F Figs). These results indicate RIG-I and TLR3 play crucial roles in restricting virus infection. Knockdown or overexpression of YAP/TAZ and LATS1/2 did not affect the expression of RIG-I and TLR7 (S5B–S5E Fig). Surprisingly, TLR3 expression was increased in YAP/TAZ- or TEADs-knockdown cells, and decreased upon LATS1/2 knockdown at both mRNA and protein levels (Fig 5A and 5D). Similarly, overexpression of YAP remarkably suppressed TLR3 expression (Fig 5B). Meanwhile, deletion of C-terminal TAD of YAP abolished YAP's ability to suppress TLR3 transcription (Fig 5B). IRF3 and p38, the downstream signaling proteins were also over-activated in YAP/TAZ- or TEADs-knockdown cells, and less activated in LATS1/2 deficient cell line (Fig 5D and 5E). Deficiency of Yap in mice also resulted in transcriptional upregulation of Tlr3 (Fig 5C), and elevated activation of Irf3 signaling in lung tissue upon infection, which was accompanied by a reduction in viral nucleocapsid (NP) protein expression upon infection (Fig 5F).

To provide further evidence that YAP/TAZ target TLR3 to participate in the regulation of the innate immune response to IAV, we performed TLR3 knockdown in YAP(5SA)-, LATS1-, as well as EV-overexpressing cells. The results showed that the phosphorylation and activation of IRF3 and p38 were predominantly suppressed by YAP(5SA) and enhanced by LATS1 in cells transfected with negative control siRNAs, meanwhile, the stimulation of IRF3 and p38 signalings upon IAV infection was substantially restrained in all EV-, YAP(5SA)- and LATS1-overexpressing cells subjected to TLR3 knockdown (Figs 5G, 5H and S6A). Consistently, exogenous TLR3 overexpression strengthened the activation of IRF3 and p38 signalings to a similar level upon knockdown of YAP/TAZ, LATS1/2 and NC (Figs 5I, 5J and S6B). Compare with siNC-knockdown cells, virus replication was no longer regulated by overexpression of YAP(5SA) and LATS1 upon TLR3 knockdown (S6C Fig). Overexpression of TLR3 also deprived the effects of YAP and LATS1 knockdown on regulation of virus replication (S6D Fig).

Additionally, NS1 decreased TLR3 expression in a dose-dependent manner, which further confirmed the role NS1 plays in antagonizing host innate immune response (S6E Fig). These above data indicate that YAP/TAZ target TLR3 pathway to modulate the innate immunity against IAV virus.

## YAP/TAZ inhibit TLR3 transcription through promotor deacetylation

We previously reported YAP/TAZ-TEADs formed a transcriptional repressor complex to silence the transcription of cyclooxygenase-2 (COX-2) [33]. Thus, we next sought to explore whether YAP/TAZ-TEADs complex also binds to the TLR3 promoter region to suppress its transcription. Two putative TEAD binding sequences (CATTCC and GGAATG) were found on the TLR3 promoter region from positions -1428 to -1423, and positions -41 to -36 respectively, referred to as P1 and P2 (Fig 6A). To assess if YAP/TAZ-TEADs directly associates with the TLR3 promoter, a chromatin immunoprecipitation (ChIP) coupled with qPCR (ChIP-qPCR) in A549 cells infected with PR8 virus or corresponding mock control was conducted. A significant recruitment of YAP was found around TEAD binding site P2. Virus infection further enhanced YAP occupancy of this region, whereas, no enrichment of YAP was found on TLR3 P1 (Fig 6B). The CYR61 promoter was used as a positive control of YAP recruitment (Fig 6B). Additionally, luciferase reporter plasmids with the TLR3 promoter and its mutants in the two putative TEAD binding sites respectively were constructed (Fig 6A). Luciferase

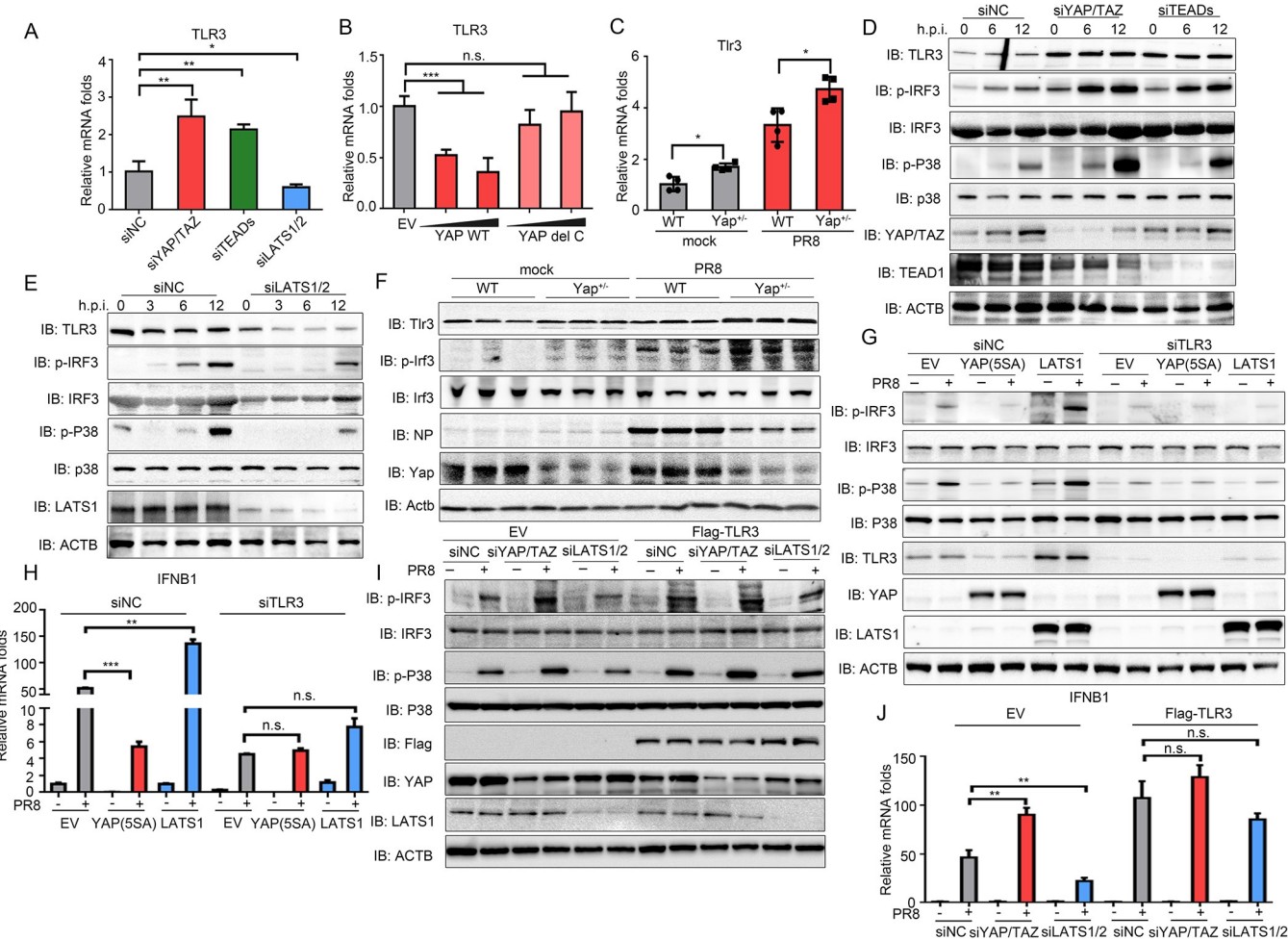

**Fig 5. YAP/TAZ inhibit TLR3-mediated innate immune responses.** (A, B) A549 cells were transfected with siRNAs targeting YAP/TAZ, TEADs and LATS1/2 (A), or transfected with plasmids of YAP and YAP del C (B). Transcriptional level of TLR3 was measured. (C) WT and Yap$^{+/-}$ mice (n = 4 per group) were infected with PR8 virus or corresponding mock control. Tlr3 expression in lung tissue was assessed by RT-qPCR. (D, E) A549 cells were infected with PR8 virus for indicated time after knockdown of YAP/TAZ, TEADs (D) or LATS1/2 (E). The cells were then harvested and subjected to immunoblotting. (F) WT or Yap$^{+/-}$ mice were infected with virus for 24 hours, and then the lungs were analyzed by immunoblotting with indicated antibodies. (G, H) EV-, YAP(5SA)-, and LATS1-stably expressing cells were transfected with siNC or siTLR3, and then infected with PR8 virus for 12 hours. Indicated protein expression levels were determined by immunoblotting (G). IFNB1 mRNA expression level was measured by RT-qPCR (H). (I, J) A549 cells stably expressing EV or TLR3 were transfected with siNC, siYAP/TAZ and siLATS1/2, and then infected with PR8 virus. Indicated protein expression levels were determined by immunoblotting (I). IFNB1 mRNA expression level was measured by RT-qPCR (J). Data in (A-B), (D-E) and (G-J) were repeated at least three times. Data in (A-C), (H) and (J) are presented as means ± SD. $^{*}p < 0.05$, $^{**}p < 0.01$, $^{***}p < 0.001$. n.s., not significant ($p > 0.05$).

reporter assays were performed by transfection of the TLR3 promoter luciferase reporter constructs together with YAP or LATS1. The transcriptional activities of WT and mutation of P1 (Mutant1) luciferase reporter genes were decreased upon YAP overexpression and elevated by overexpression of LATS1 (Fig 6C and 6D). In contrast, mutation of P2 (Mutant2) did not show the change of luciferase activity in a YAP or LATS1-dependent manner, indicating this putative TEADs binding site is indispensable for the association of YAP/TAZ with the TLR3 promoter (Fig 6C and 6D).

Acetylation and methylation of histone proteins are two major types of epigenetic modifications that regulate gene expression [37]. Our previous finding demonstrated that YAP/TAZ associate with histone deacetylases (HDACs) family member HDAC7 to suppress COX-2 expression [33]. To explore whether HDACs or methyltransferases are involved in blockage of

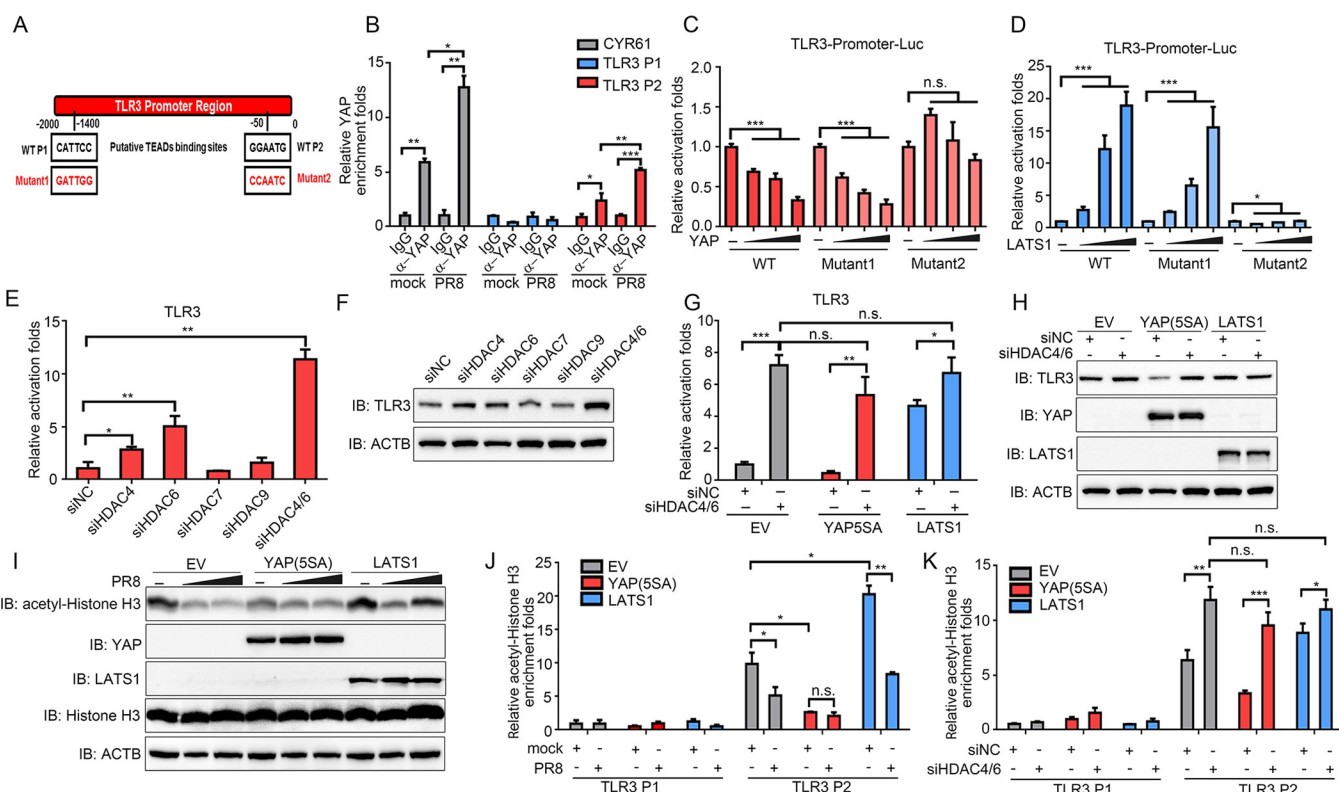

**Fig 6. YAP/TAZ promote histone deacetylation in the TLR3 promoter region via HDAC4/6.** (A) Schematic representation of the TLR3 promoter with the putative TEADs binding sites was shown. Corresponding mutation sites constructed in the luciferase plasmids were presented below. (B) A549 cells were infected with PR8 virus at an MOI of 0.01 or mock control for 12 hours. ChIP assay was then conducted using anti-YAP or normal rabbit IgG. Precipitated DNA was measured by qPCR using primers to amplify the indicated regions surrounding the putative TEAD-binding sites. (C, D) Luciferase reporter activities driven by wild type or mutants of the TLR3 promoters with overexpression of YAP (C) or LATS1 (D) were assessed. (E, F) A549 cells were transfected with indicated siHDACs. The expression of TLR3 was determined by RT-qPCR (E) and immunoblotting (F). (G, H) siNC or siHDAC4/6 was transfected into EV, YAP(5SA) and LATS1-overexpressing cells. TLR3 expression was measured by RT-qPCR (G) and immunoblotting (H). (I) Cells overexpressing of EV, YAP (5SA) and LATS1 were infected with mock control or PR8 virus at an MOI of 0.005 and 0.01 for 24 hours. Indicated protein expression levels were measured by immunoblotting. (J, K) EV, YAP(5SA) and LATS1-overexpressing cells were infected with PR8 virus at an MOI of 0.01 or mock control for 12 hours (J), or transfected with siNC and siHDAC4/6 (K). ChIP assay was then conducted using anti-acetyl-Histone H3 antibody or normal rabbit IgG. Precipitated DNA was measured by qPCR using primers to amplify the indicated regions on the TLR3 promoter. All experiments were repeated at least three times. Data in (B-E), (G) and (J-K) are presented as means ± SD. $^*p < 0.05$, $^{**}p < 0.01$, $^{***}p < 0.001$. n.s., not significant ($p > 0.05$).

TLR3 transcription by YAP/TAZ, we treated A549 cells with Trichostatin A (TSA), a HDACs inhibitor involved in chromatin remodeling [38], or UNC0638, a selective methyltransferase inhibitor for G9a (EHMT2) and GLP (EHMT1) histone methyltransferase [39]. We found both mRNA and protein levels of TLR3 were induced by TSA (S7A and S7B Fig). Meanwhile, UNC0638 showed no effect on TLR3 expression (S7C and S7D Fig). Furthermore, YAP/TAZ-hampered TLR3 transcription was partially reversed by TSA treatment, meanwhile LATS1-enhanced TLR3 expression was further increased upon TSA treatment, demonstrating HDACs may participate in the regulation of TLR3 transcription by YAP/TAZ (S7E and S7F Fig). On the other side, knockdown of YAP/TAZ disrupted the suppressive effect of TSA on TLR3, while knockdown of LATS1/2 did not, which implied YAP/TAZ may be indispensable for the suppressive function of HDACs on the TLR3 promoter region (S7G and S7H Fig). The above results suggest that YAP/TAZ may recruit HDACs to modulate TLR3 transcription.

Subsequently, we set out to determine which HDAC mediates the suppressing function of YAP/TAZ on TLR3 transcription. Knockdown of HDAC4/6/7/9 in A549 cells demonstrated loss of HDAC4 and HDAC6 both enhanced the expression of TLR3. Knockdown of HDAC4/

6 at the same time further elevated TLR3 transcription, implying they functioned as suppressors of TLR3 transcription parallelly (Fig 6E and 6F). Moreover, disruption of HDAC4/6 in YAP(5SA)-, LATS1- and EV-overexpressing cells boosted TLR3 expression to a comparable level, confirming the critical role HDAC4/6 play in mediating the repression function of YAP/TAZ (Fig 6G and 6H).

We then queried whether changes in the histone acetylation state occurred when cells were infected with IAV. Intriguingly, we found the global acetylation of histone H3 at lysines 9, 14, 18, 23 and 27 in cells was substantially diminished upon IAV infection. Meanwhile, YAP(5SA) hampered, and LATS1 enhanced the acetylation of histone H3 (Fig 6I). We next utilized ChIP-qPCR to investigate how the acetylated histone H3 chromatin occupancy in the TLR3 promoter region be regulated by YAP/TAZ upon infection. This analysis revealed that the acetylated histone H3 occupancy in the TLR3 promoter P2 site was significantly reduced after PR8 virus infection in both EV- and LATS1-overexpressing cells, but not in YAP(5SA)-overexpressing cells. Besides, compared to EV-overexpressing cells, LATS1 increased acetylated histone H3 abundance in P2 site of the TLR3 promoter. Meanwhile, YAP(5SA) overexpression resulted in reduced acetylated histone H3 abundance to a level similar in mock-infected and virus-infected cells (Fig 6J). In contrast, the acetylated histone H3 was barely accumulated in P1 site of TLR3 promoter, which indicated P1 site may not bind with the acetylated histone H3 (Fig 6J). In addition, we found loss of HDAC4/6 promoted acetylated histone H3 abundance in the TLR3 promoter region to a comparable level in EV-, YAP(5SA) and LATS1-overexpressing cells, which indicated HDAC4/6 predominantly mediated the YAP/TAZ-regulated histone H3 acetylation state in the TLR3 promoter region (Fig 6K). Together, these results suggest the YAP/TAZ-diminished acetylation of histone H3 through HDAC4/6 regulates the transcription of TLR3.

## Discussion

It is well documented that the Hippo pathway is tightly associated with many human cancers, and its main downstream effector proteins YAP/TAZ play key roles in tumorigenesis [40,41]. Thus, recent findings associating Hippo pathway with virus-induced diseases, including hepatitis B virus (HBV) and human papillomavirus (HPV) infection are intriguing [42,43,44,45]. Mechanistically, some hypothesized that viral proteins such as HPV E6 and HBV X protein (HBx) could activate YAP/TAZ [42,45]. A recent report also showed YAP is activated upon activation of IRF3 antiviral signaling by viral infection [46]. Although, previous studies have identified plenty of host genes could facilitate IAV replication [47,48], our study shows for the first time a reciprocal regulation between YAP/TAZ and IAV. Multifunctional protein of IAV NS1 activates the activity of YAP/TAZ. Meanwhile, YAP/TAZ antagonize the innate immune response through TLR3 transcriptional silencing, favoring viral replication (Fig 7).

Some other studies showed YAP/TAZ are negative regulators of the innate immunity against viruses. However, the underlying mechanisms of the suppression functions are varied. YAP/TAZ were reported to inhibit the innate antiviral defense by preventing the K63-linked ubiquitination of TBK1 and disrupting its interaction with MAVS, STING and IRF3 [29]. Previous work also suggested YAP interacts with IRF3, preventing its dimerization and nuclear translocation, which further reduces the production of IFN-β and ISGs in response to virus infection [30]. In our work, YAP/TAZ suppress the innate immune response against IAV through silencing the transcription of TLR3. We provide evidences that either knockdown or exogenously expression of TLR3 could tremendously abrogate the regulation of type I IFN and inflammatory signalings by YAP/TAZ, which indicated that TLR3 is a crucial target for YAP/TAZ to tightly regulate the innate immune response during IAV infection (Figs 5G–5J and

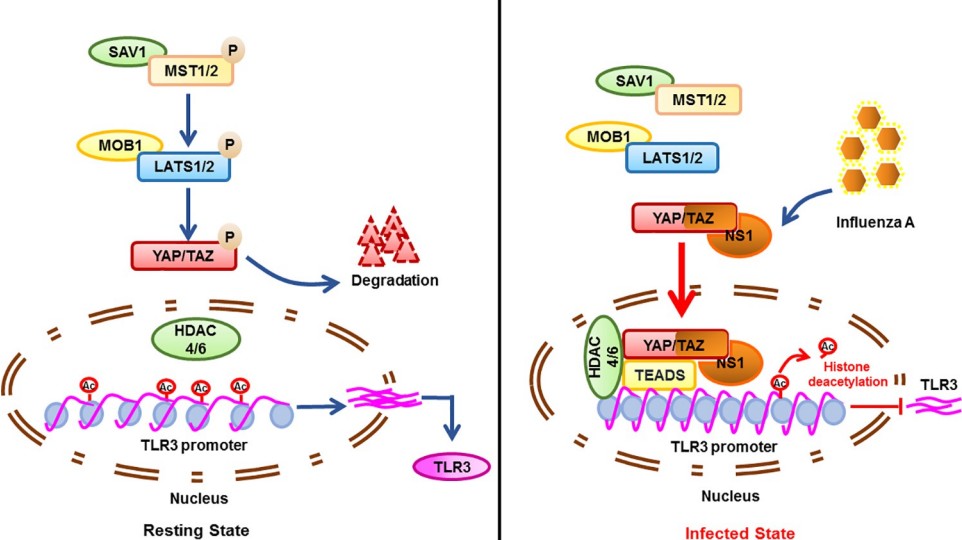

**Fig 7. Molecular model depicts the mechanism of YAP/TAZ to antagonize TLR3-mediated innate immune signaling upon IAV infection.** When the Hippo signaling is on, YAP/TAZ are restricted to the cytoplasm and targeted for degradation. TLR3 is consistently transcribed. Conversely, IAV infection leads the activation of YAP/TAZ and nuclear localization through direct protein-protein interactions between NS1 and the C-terminal TAD of YAP. YAP/TAZ-TEADs complex functions as a transcriptional repressor to suppress the transcription of TLR3 by histone deacetylation at the specific promoter region through HDAC4/6.

S6A–S6D). Our study provides a novel insight into the mechanism underlying the regulation of the innate immune signaling by YAP/TAZ.

Our study suggests YAP/TAZ could bind to the TLR3 promoter and diminish its acetylated histone H3 occupancy, thus inhibiting its transcription (Fig 6B–6D and 6J). This finding is in agreement with previous reports [33,49,50,51]. Of note, Beyer *et al.* and Kim *et al.* both suggests that nucleosome remodeling and deacetylation (NuRD) complex mediates the repression function of YAP [49,50,52]. In parallel, our previous study also showed that histone deacetylase HDAC7 is recruited by YAP/TAZ-TEADs complex to exert its transcriptional inhibition function [33,53]. In our present study, YAP/TAZ utilize HDAC4/6 to deacetylate histone H3 in the promoter region of TLR3, and silence its transcription (Fig 6E–6H and 6K). Together, these studies imply that as the well-known transcriptional co-activators, YAP/TAZ have significant transcriptional repression functions under relevant physiologic conditions. However, the exact targets and the underlying molecular mechanism require further investigation.

Studies have shown that influenza virus is recognized by different PRRs, including RIG-I, TLR3 and TLR7. However, in this study, we found RIG-I and TLR3, but not TLR7, play important roles in restricting virus replication upon IAV infection in A549 cells (S5A Fig), which was also supported by previous report that RIG-I and TLR3, but not TLR7 are responsible for IFN induction by influenza virus in human lung alveolar epithelial cells [54]. More work is needed in the future to address whether TLR7 is needed for triggering IFN signaling in other cell types in the lung during IAV infection. It is also worth noting that the role of TLR3, which senses the dsRNA in the endosomes in host response to influenza virus is controversial. Although some suggested that dsRNA may be not generated during influenza virus replication [6,55]. It was reported that TLR3 contributes directly to the immune response of respiratory epithelial cells to IAV, and stimulates the production of pro-inflammatory and anti-viral cytokines [56,57], whereas, upon lethal influenza virus infection, Tlr3$^{-/-}$ mice display reduced inflammatory mediators and survive longer than WT mice, despite sustaining higher viral

production in the lungs [58,59]. A previous report claimed sensing of IAV by TLR3 primarily regulates proinflammatory response, but not type I IFN-dependent antiviral signaling [58]. Our study provides evidence that TLR3 drives both the activation of IRF3-mediated type I IFN-dependent antiviral signaling and P38-mediated proinflammatory signaling in host response to IAV (Figs 5G–5J, S6A and S6B), and markedly restricted virus replication (S6C and S6D Fig). Hence, TLR3 could be an important molecular target to be investigated for management of IAV infections.

The mechanisms of innate immune responses to IAV infection and viral immune evasion mechanisms have been intensively researched for the past decades. The IAV multifunctional protein NS1 is a well-known virulence factor that obstructs the innate immune response and enhances virus pathogenicity. However, to our knowledge, the mechanism of NS1 to inhibit host innate immunity via the Hippo signaling effectors YAP/TAZ has never been studied before. Based on our study, NS1 activates YAP/TAZ by direct interaction with YAP/TAZ through their C-terminal TAD (Fig 2D and 2E). This is further supported by the observation LATS1/2 activities are not altered in the context of infection (Fig 1A). Given that NS1 is capable of nuclear translocation [60,61], we reason that NS1 may facilitate nuclear targeting of YAP/TAZ for activation. It will be tempting to explore a more detailed molecular basis such as the specific sites or motifs needed for their binding.

Overall, the work presented in this study demonstrates that IAV multifunctional protein NS1 hijacks cellular YAP/TAZ to facilitate its antagonism of the host antiviral response via suppressing TLR3 signaling. This work further strengthens our mechanistic understanding of how IAV evades the host antiviral response, and how the Hippo signaling modulates the innate immune signaling, exposing a new potential therapeutic target against viral infection.

## Materials and methods

### Ethics statement

All animal experiments were approved by the Animal Care and Use Committees of The First Affiliated Hospital, School of Medicine, Zhejiang University (2021–6)

### Mice and cell lines

Yap deficient mice on a C57BL/6J background were generated using CRISPR-Cas9 system by GemPharmatech. Its littermates with wild type genotype were used as controls.

The A549, HEK293T and MDCK cell lines were obtained from American Type Culture Collection (ATCC). Cells were maintained in Dulbecco's Modified Eagle's Medium supplemented with 10% FBS (Gibco) and 1% penicillin-streptomycin (HyClone). Chemical reagents TSA and UNC0638 were purchased from Selleck Chemicals. Poly I:C was purchased from Sigma.

### Virus propagation and infection

Influenza A virus (H1N1) strain A/PR/8/34 (PR8) and PR8-GFP were propagated in the allantoic cavity of 10-day-old embryonated specific-pathogen-free chicken eggs, and viral titers were determined by standard plaque assay on confluent monolayers of MDCK cells as previously described [62]. A549 cells were infected with PR8 virus at the given MOI and media only, used as mock control. After one hour of adsorption, the viral inoculum was discarded and replaced by Opti-MEM added with 0.5–1 μg/ml TPCK-treated trypsin (Thermo Fisher Scientific) in a 37˚C incubator. Cells were harvested at select time points for different assays. Viral titers were determined by standard plaque assay on confluent monolayers of MDCK cells.

For mice infection, six-week-old female wild type, or Yap$^{+/-}$ C57BL/6J mice were anesthetized with pentobarbital prior to intranasal infection with $10^3$ PFU of PR8 virus in 50 ml of PBS (Gibco) or mock infected. Lungs were harvested at indicated timepoints and homogenized for analysis. Mice were monitored daily for weight loss over a 5-day period.

### Quantitative RT-PCR assay

Cells were lysed, and total RNA extracted using TRIzol (TaKaRa) according to the manufacturer's instructions. cDNA was generated by HiScript II RT SuperMix (Vazyme). Quantitative real-time PCR was performed using SYBR qPCR Master Mix (Vazyme) and QuantStudio 5 real-time PCR system (Applied Biosystems). The primer sequences are provided in S1 Table.

### RNA interference and plasmids

The siRNA specific to indicated genes and silencer negative control were transfected into A549 cells using Lipofectamine 2000 Transfection Reagent (Thermo Fisher Scientific) according to the manufacturer's instructions. The details of siRNA sequences are provided in S1 Table. YAP, TAZ, YAP(5SA), YAP del C and LATS1 were all constructed in pLX304-puro vector. NS1 of the different IAV strains were constructed into c-Flag pcDNA3.1 vector. The CTGF promoter, CTGF ΔTB mutant and synthetic TEAD luciferase reporter GTIIC were cloned into pGL3-Basic vector as previously described [63,64]. Mutant1 and Mutant2 of the TLR3 promoters were generated by mutating CATTCC site to GATTGG and GGAATG site to CCAATC respectively. Mutant1 and Mutant2, together with WT were cloned into pGL4.23 vector. Plasmids were transiently transfected into cells with Lipofectamine 2000 Transfection Reagent. For stable transfection, A549 cells were then selected with puromycin (1 μg/ml) and pooled for further experiments.

### Luciferase reporter assay

A549 cells were transfected with CTGF or GTIIC reporters along with the pRL-Luc with Renilla luciferase coding as the internal control for transfection and other expression plasmids, or infected with virus. In brief, after 24 hours of transfection and infection, cells were lysed by passive lysis buffer (Promega). Luciferase assays were performed using a dual luciferase assay kit (Promega), quantified with GloMax 20/20 Luminometer (Promega).

### ChIP assay

ChIP assay was performed as previously reported [33]. Oligo sequences used in ChIP assays are available in S1 Table. Antibody information is provided in S2 Table.

### Immunofluorescence, microscopy and analysis

A549 cells grown on glass coverslips and infected with mock media or PR8 virus as indicated. Cells were then washed with PBS and fixed with 4% paraformaldehyde in PBS, permeabilized with 0.2% Triton X-100 (Sigma-Aldrich), blocked with 3% bovine serum albumin (Sangon Biotech) and incubated sequentially with Yap antibody and Alexa Fluor–labeled secondary antibodies with extensive washing. Nuclei was counterstained with DAPI. Immunofluorescence images were obtained using Olympus FV3000 confocal microscope. For visualization of PR8-GFP replication in cells, the ImageXpress Pico (Molecular Devices) was used to visualize and quantify the mean fluorescence intensity of each cell. Detailed antibody information is provided in S2 Table.

### Isolation of nuclear and cytoplasmic protein

A549 cells were infected with PR8 virus for indicted time. Nuclear and cytoplasmic proteins were isolated using Nuclear and Cytoplasmic Protein Extraction kit following the detailed instructions provided (Beyotime). Cell fractions were then analyzed by immunoblotting for expression of YAP/TAZ, GAPDH and Histone3.

### Annexin V-FITC/propidium iodide (PI) apoptosis assay

A549 cells stably expressing YAP(5SA), LATS1 or EV were infected with PR8 virus for 12 hours before stained with Annexin V-FITC and PI according to the manufacturer's protocol (Annexin V-FITC/PI Apoptosis Assay kit, BD Pharmingen). Frequency of apoptotic cells was measured by flow cytometry (C6; BD Biosciences). Flow cytometry data were analyzed using FlowJo.

### Coimmunoprecipitation and immunoblot analysis

RIPA buffer with cocktail protease inhibitors (Selleck Chemicals) was used for cells lysis and tissues. For coimmunoprecipitation studies, HEK293T cells transfected with the indicated plasmids were lysed with lysis buffer (20 mM Tris-HCl, pH 7.4, 150 mM NaCl, 0.5% Nonidet P-40, 10% glycerol, 1 mM DTT, and complete protease inhibitor mixture), and subjected to immunoprecipitation using M2-conjugated magnetic beads (Sigma). M2 beads were resolved in 2XSDS loading buffer and analyzed by SDS-PAGE and immunoblotting with the indicated antibodies and secondary anti-mouse or anti-rabbit antibodies conjugated to horseradish peroxidase (HRP). Detailed information of all antibodies used in immunoblotting analysis is provided in S2 Table. Chemiluminescence was acquired on ChemiScope 3300 Pro (Clinx).

### Enzyme-linked immunosorbent assay (ELISA)

The concentrations of IFN-β in mice lung tissue were measured with commercial ELISA Kits (R&D Systems). Briefly, lungs from mock or virus-infected mice were homogenized in 1 ml of PBS using a homogenizer. A seven-point standard curve was prepared using the manufacturer's protocol, and interferon titers in the samples were determined in reference to the standards using a 4-parameter fit. Optical densities were read by Epoch 2 Microplate Spectrophotometer (BioTek) at an absorbance of 450 nm, normalized by 540 nm subtraction for optical imperfections correction.

### Hemotoxylin and Eosin (H&E) staining and TUNEL assay

The lung tissue from mice was soaked in 10% formalin, dehydrated through graded alcohols and embedded in paraffin wax. The sections were then cut into slices from these paraffin-embedded tissue blocks. The slices were deparaffinized by immersing in xylene and rehydrated. The slices were dyed with hematoxylin and eosin, and then rinsed with water. Each slide was dehydrated through graded alcohols. Lung sections were soaked in xylene twice. To evaluate apoptosis in the lung tissue sections, TUNEL staining was performed by using CF640 Tunel Cell Apoptosis Detection Kit (Servicebio) according to the manufacturer's protocol. The TUNEL intensity was examined under a fluorescence microscope (IX83, Olympus). The percentage of TUNEL-positive cells was calculated by ImageJ.

### Quantification and statistical analysis

Statistical analysis was performed with GraphPad Prism 6. P values were determined by unpaired two-tailed Student's t-test (for comparing two groups), or one-way ANOVA (for

comparing more than two groups) in cell-based assays. Statistical differences in mice-based assays were calculated using the Mann-Whitney U test. Statistical analyses of body weight loss curves were performed with two-way ANOVA for multiple comparisons. All analyzed data are expressed as mean ± standard deviation (SD). Differences with a *p* value < 0.05 were considered significant.

## Supporting information

**S1 Fig. Il-6 and Ifnb1 are induced upon virus infection.** (A) Female C57BL/6J mice (n = 6 per group) were infected intranasally with $10^3$ PFU of PR8. Lungs were harvested at different days post inoculation for analyzing changes in mRNA expression of Il-6 and Ifnb1. Data are presented as means ± SD. *p < 0.05, **p < 0.01, ***p < 0.001.
(TIF)

**S2 Fig. NS1 interacts with YAP/TAZ.** (A) 293T cells were co-transfected with TAZ or YAP plasmids, with Flag-EV and Flag-NS1 of the indicated IAV strains. WCL was subjected to immunoprecipitation with M2 beads. Bound proteins and input lysis were analyzed by immunoblotting with indicated antibodies. All of the experiments were repeated at least three times.
(TIF)

**S3 Fig. The efficiencies of plasmids and siRNAs transfection in A549 cells are presented.** (A) The expression levels of stably expressed exogenous YAP(5SA) and LATS1 were determined by immunoblotting. (B-F) The changes in mRNA expressions of YAP/TAZ (B), LATS1/2 (C), TEAD1 (D), RIG-I (E) and TLR7 (F) in A549 cells transfected with corresponding siRNAs were analyzed by RT-qPCR. All of the experiments were repeated at least three times. Data in (B-F) are presented as means ± SD. *p < 0.05, **p < 0.01, ***p < 0.001.
(TIF)

**S4 Fig. The Hippo signaling controls replication of IAV both *in vitro* and *in vivo*.** (A, B) Virus copies from cells transfected with indicated plasmids (A) or siRNAs (B) were determined by RT-qPCR. (C, D) The expression of Yap (C) and virus copies (D) in PR8-infected WT and Yap+/- mice (n = 4 per group) were assessed by RT-qPCR. Data in (A, B) were repeated at least three times. Data are presented as means ± SD. *p < 0.05, **p < 0.01, ***p < 0.001.
(TIF)

**S5 Fig. The expression levels of RIG-I and TLR7 are not regulated by YAP/TAZ or LATS1/ 2.** (A) A549 cells transfected with siNC, siRIG-I, siTLR3 and siTLR7 were infected with PR8-GFP. GFP fluorescence was detected (left). Cell average fluorescence intensity per cell was calculated (right). (B, C) The detection of indicated protein levels was conducted using immunoblotting in A549 cells with knockdown of YAP/TAZ and LATS1/2 (B), or overexpression of YAP(5SA) and LATS1 (C). (D, E) The mRNA expressions of RIG-I (D) and TLR7 (E) were detected upon YAP/TAZ and LATS1/2 knockdown, or YAP(5SA) and LATS1 overexpression. All of the experiments were repeated at least three times. Data in (A, D-E) are presented as means ± SD. *p < 0.05, **p < 0.01, ***p < 0.001. n.s., not significant (p > 0.05).
(TIF)

**S6 Fig. TLR3 mediates the Hippo-regulated replication of IAV in host cells.** (A) EV-, YAP (5SA)-, and LATS1-overexpressing cells were transfected with siNC or siTLR3, and then infected with PR8 virus for 12 hours. IRF7 and IFIT1 expression levels were assessed by RT-qPCR. (B) siNC, siYAP/TAZ and siLATS1/2 were transfected into cells stably expressing EV or TLR3. PR8 was added to cells for 12 hours before IRF7 and IFIT1 expression levels were

assessed by RT-qPCR. (C) EV-, YAP(5SA)-, and LATS1-overexpressing cells were transfected with siNC or siTLR3. Virus copies were measured. (D) Cells stably expressing TLR3 were transfected with NC scrambled siRNA and siRNAs targeting YAP/TAZ or LATS1/2. PR8 was added to cells for 12 hours before virus copies were measured. (E) Different amounts of Flag-NS1 were transfected into A549 cells. WCL was subjected to RT-qPCR (left) and immunoblotting (right) for measurement of TLR3 expression. All of the experiments were repeated at least three times. Data in (A-E) are presented as means ± SD. *p < 0.05, **p < 0.01, ***p < 0.001. n. s., not significant (p>0.05).
(TIF)

**S7 Fig. TSA restores the expression of TLR3 regulated by YAP/TAZ.** (A, B) Indicated concentrations of TSA were added to cells. Cells were then harvested and RT-qPCR for mRNA level (A) and immunoblotting for protein level (B) of TLR3 were conducted. (C, D) Cells were treated with UNC0638 before lysed for detection of mRNA level (C) and protein level (D) of TLR3. (E, F) A549 cells were transfected with YAP(5SA) and LATS1 (E) for 24 hours. Cells were then treated with TSA (100 and 500 nM) for another 6 hours before RT-qPCR analysis (E) and immunoblotting (F). (G, H) siNC, siYAP/TAZ and siLATS1/2 were transfected into A549 cells. TSA were treated before harvesting for RT-qPCR analysis (G) and immunoblotting (H). All of the experiments were repeated at least three times. Data in (A, C, E and G) are presented as means ± SD. *p < 0.05, **p < 0.01, ***p < 0.001. n.s., not significant (p>0.05).
(TIF)

**S1 Table. Primers&siRNAs used in this study.**
(DOCX)

**S2 Table. Antibodies used in this study.**
(DOCX)

## Author Contributions

**Conceptualization:** Qiong Zhang, Hongyan Diao.

**Formal analysis:** Qiong Zhang, Xujun Zhang.

**Funding acquisition:** Qiong Zhang, Hai Wang, Hongyan Diao.

**Investigation:** Qiong Zhang, Xujun Zhang, Jingjing Jiang, Yuchong Wang, Kefan Bi.

**Project administration:** Xiaobo Lei, Hongyan Diao.

**Resources:** Xiaobo Lei, Hai Wang.

**Supervision:** Xiaobo Lei, Hongyan Diao.

**Writing – original draft:** Qiong Zhang.

**Writing – review & editing:** Qiong Zhang, Xiaobo Lei, Hongyan Diao.

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
