## [Decision Letter · Decision Letter 0]

3 Jan 2022

Dear Dr. Diao,

Thank you very much for submitting your manuscript "Influenza A virus NS1 protein hijacks YAP/TAZ to suppress TLR3-mediated innate immune response" for consideration at PLOS Pathogens. As with all papers reviewed by the journal, your manuscript was reviewed by members of the editorial board and by several independent reviewers. In light of the reviews (below this email), we would like to invite the resubmission of a significantly-revised version that takes into account the reviewers' comments. The reviewers note the interest in TLR3-mediated responses during influenza virus infection and the novel finding that NS1 may regulate TLR3 via YAP/TAZ. However, the reviewers also highlight points that should be addressed to enhance confidence in the observations and the generalities of their conclusions. Please pay particular attention to the comments below as you revise your submission. 

We cannot make any decision about publication until we have seen the revised manuscript and your response to the reviewers' comments. Your revised manuscript is also likely to be sent to reviewers for further evaluation.

Sincerely,

Andrew Mehle

Pearls Editor

PLOS Pathogens

Sonja Best

Section Editor

PLOS Pathogens

Kasturi Haldar

Editor-in-Chief

PLOS Pathogens

orcid.org/0000-0001-5065-158X

Michael Malim

Editor-in-Chief

PLOS Pathogens

orcid.org/0000-0002-7699-2064

Reviewer's Responses to Questions

**Part I - Summary**

Reviewer #1: Zhang et al. report a new finding that IAV NS1 activates YAP/TAZ through direct binding, suppressing TLR3 expression. The authors also indicated that activated YAP/TAZ suppressed the function of HDACs.

The authors provided solid evidence for the following;

1. IAV NS1 induces activation of YAP/TAZ

2. Activated YAP/TAZ is co-translocated to the nucleus by NS1 upon IAV infection

3. NS1 binds to the C-terminal TAD of YAP

4. Activated YAP/TAZ antagonize the TLR3 transcription, leading to enhanced viral replication.

To this reviewer, it seems that this study was carefully performed. The manuscript was well-written and easy to follow. The authors’ findings are very interesting and important for understanding the role of IAV NS1 in antagonizing immune responses. However, a minor revision will be necessary to address the following concerns.

Reviewer #2: In this manuscript the authors discuss the immunomodulatory role of the Hippo signaling pathway in IAV infection. Although several studies over the past decade have underscored the importance of the RIG-I signaling pathway, and mechanistically defined its regulation, very few studies have focused on the TLR3-mediated regulation of immune responses during flu infection. This study is therefore an important contribution to the field. General and specific comments on the article are provided below:

**Part II – Major Issues: Key Experiments Required for Acceptance**

Reviewer #1: Major questions :

1. The authors observed the IRF3 activation upon the activation of YAP (Fig 1F). This seems contradictory to the observation that deficiency of YAP resulted in elevated IRF3 activation (Fig 5F). If NS1 inhibited the expression of IRF3, why is IRF3 activated upon IAV infection?

2. To avoid unnecessary confusion, the authors need to describe the protein constructs more accurately. For example, the residue numbers in YAP del C should be described (i.e., residues xx-xx). Other proteins should also be described the same way.

3. Ref 36 indicated that the C-terminal TAD of YAP is responsible for binding to HDAC7. Somehow, the authors did not connect the present results with the previous finding. For example, ref 36 should be mentioned in line 349, although they cited the paper later in the discussion section.

4. Similarly, the authors did not cite ref 36 when describing their finding that NS1 binds to the C-terminal TAD of YAP (lines 184-192). This makes it difficult to understand the rationale of their test; i.e., why did the authors suddenly test the binding of NS1 and the C-terminal TAD? Citing the paper in the right place will help readers follow the authors’ rationale behind their experimental design.

5. Activation of YAP/TAZ can antagonize the function of TBK1 and IRF3. The authors also mentioned these previous findings. However, it is unclear whether the authors indicate that suppressing TLR3 transcription is the dominant mechanism by which NS1 interferes with immune responses. So, the authors need to clarify in the discussion section if they exclude other mechanisms.

Reviewer #2: 

General comments:

For all of the experiments, the authors have used the mouse adapted flu strain PR8 to show the effect of NS1 on the effector proteins of the Hippo pathway. Although PR8 serves as an appropriate model (especially since the study uses mouse models) it would be useful to use other strains of flu to corroborate whether these findings are universal in flu infection. At least in the transfection experiments, NS1 from other flu strains should be included to check for binding to YAP/TAZ to confirm universal phenotypes. 

Specific comments:

In Fig 3, the authors indicate increased production of flu virus in cells constitutively expressing active YAP. This needs to be confirmed by measuring viral titres using plaque assays along with fluorescence measurements that the authors have performed. Increase in fluorescent punctae may not necessarily lead to production of infectious virions and it would be useful to determine whether this is the case or defective particles are produced. With the current set of experiments it is not clear which step in the viral lifecycle is facilitated by YAP/TAZ dependent immune regulation e.g. entry, replication, assembly or release.

Along the same lines, it is difficult to understand the magnitude of the effect in Fig 3B; there seems to be almost a complete block in virus production in the YAP-depleted cells based on the fluorescence data. From previous literature, the TLR3 signaling is thought to play a minor role in restricting infection compared to RIG-I. The authors should add some discussion on this point. It might be useful to have RIG-I ko/overexpressing cells as controls in this experiment. It would also be useful to check whether other pathways are affected, such as TLR7, to be able to quantify the magnitude of this phenotype.

In Fig 4, dsRNA or Poly I:C controls should be included in the siRNA knockdown experiments to measure IFN and other cytokine responses to delineate effects of other viral components that might perturb this pathway. Transcriptional expression/repression of target genes should be recapitulated in these controls.

In Fig 5, the increase in immune signaling proteins is clear from the YAP dose-dependent experiment. It is however not clear that this arises from TLR3 signaling. Expression of other signaling pathways e.g. TLR7 should also be measured. Does KD/overexpression of TLR3 correlate negatively with YAP phenotype?

**Part III – Minor Issues: Editorial and Data Presentation Modifications**

Reviewer #1: Minor points

1. Lines 166 and 167; Figure 1E and Fig 1F should be swapped.

2. Some references are not correctly formatted. For example, see ref 30.

Reviewer #2: There are also several grammatical and typographical errors in the current version of the manuscript, which needs to be corrected.

PLOS authors have the option to publish the peer review history of their article (what does this mean?). If published, this will include your full peer review and any attached files.

Reviewer #1: No

Reviewer #2: **Yes: **Sumana Sanyal
---

## [Decision Letter · Decision Letter 1]

6 Apr 2022

Dear Dr. Diao,

We are pleased to inform you that your manuscript 'Influenza A virus NS1 protein hijacks YAP/TAZ to suppress TLR3-mediated innate immune response' has been provisionally accepted for publication in PLOS Pathogens.

Before your manuscript can be formally accepted you will need to complete some formatting changes, which you will receive in a follow up email. A member of our team will be in touch with a set of requests. You will also see reviewer 2 has made a minor suggestion about a point to add to the discussion. It is your decision whether you want to included this suggested comment, and it is not needed for acceptance.

Best regards,

Andrew Mehle

Pearls Editor

PLOS Pathogens

Sonja Best

Section Editor

PLOS Pathogens

Kasturi Haldar

Editor-in-Chief

PLOS Pathogens

orcid.org/0000-0001-5065-158X

Michael Malim

Editor-in-Chief

PLOS Pathogens

orcid.org/0000-0002-7699-2064

Reviewer Comments (if any, and for reference):

Reviewer's Responses to Questions

**Part I - Summary**

Reviewer #1: The authors nicely addressed all my concerns about the original manuscript.

Reviewer #2: The authors have addressed all the concerns previously raised. I have no additional concerns regarding this study. The authors could consider including a line in the discussion that deltaNS1 IAV is expected to not trigger YAP/Taz dependent suppression of TLR3 signalling.

**Part II – Major Issues: Key Experiments Required for Acceptance**

Reviewer #1: No major issues found.

Reviewer #2: No further issues

**Part III – Minor Issues: Editorial and Data Presentation Modifications**

Reviewer #1: No minor issues found.

Reviewer #2: No further issues

PLOS authors have the option to publish the peer review history of their article (what does this mean?). If published, this will include your full peer review and any attached files.

Reviewer #1: No

Reviewer #2: **Yes: **Sumana Sanyal

---

## [Editor Report · Acceptance letter]

28 Apr 2022

Dear Dr. Diao,

We are delighted to inform you that your manuscript, "Influenza A virus NS1 protein hijacks YAP/TAZ to suppress TLR3-mediated innate immune response," has been formally accepted for publication in PLOS Pathogens.

Best regards,

Kasturi Haldar

Editor-in-Chief

PLOS Pathogens

orcid.org/0000-0001-5065-158X

Michael Malim

Editor-in-Chief

PLOS Pathogens

orcid.org/0000-0002-7699-2064